# Adaptations of the Autonomic Nervous System and Body Composition After 8 Weeks of Specific Training and Nutritional Re-Education in Amateur Muay Thai Fighters: A Clinical Trial

**DOI:** 10.3390/sports13030072

**Published:** 2025-03-03

**Authors:** Antonio Beira de Andrade Junior, Elena Marie Peixoto Ruthes de Andrade, Guilherme Rodrigues de Souza, Agnaldo José Lopes

**Affiliations:** 1Rehabilitation Sciences Postgraduate Program, Centro Universitário Augusto Motta (UNISUAM), Rua Dona Isabel, 94, Bonsucesso, Rio de Janeiro 21032-060, Brazil; antoniobeira96@gmail.com; 2Postgraduate Program in Health Sciences, Universidade do Sul de Santa Catarina (UNISUL), Avenida José Acácio Moreira, 787, Dehon, Tubarão 88704-001, Brazil; epeixotoruthes@gmail.com; 3Physical Education Course, Centro Universitário Campos de Andrade (UNIANDRADE), Rua João Scuissiato, 001, Santa Quitéria 80310-310, Brazil; guilhermeoffrodrigues@gmail.com

**Keywords:** Muay Thai, autonomic nervous system, body composition, intervention

## Abstract

Background: Considering that the nervous system regulates cardiac autonomic modulation (CAM) and that low CAM is associated with poorer performance, it is essential to evaluate the effects of training to increase parasympathetic modulation in Muay Thai (MT) fighters. Therefore, the aim of this study was to evaluate the effects of an 8-week intervention based on strength training and nutritional counseling on performance, CAM, and nutritional status in amateur MT fighters. Methods: This is a longitudinal and interventional study in which 22 MT fighters underwent a strength training program and nutritional protocol. Before and after the intervention, they underwent the ten-second frequency speed of kick test (FSKT-10s), multiple frequency speed of kick test (FSKT-mult), bioimpedance analysis (BIA), and assessment of heart rate variability. Results: After the intervention, there was an increase in the number of kicks in both FSKT-10s and FSKT-mult (*p* = 0.0008 and *p* = 0.032, respectively). In BIA, there was a significant increase in both fat-free mass and basal metabolic rate (*p* = 0.031 and *p* = 0.020, respectively). After the intervention, significant increases were observed during the physical test in the following variables that denote improvement in parasympathetic modulation: square root of the mean squared differences of successive RR intervals (*p* = 0.005); percentage of adjacent RR intervals with a difference in duration greater than 50 ms (*p* = 0.002); high frequency range (*p* < 0.0001); and standard deviation measuring the dispersion of points in the plot perpendicular to the line of identity (*p* = 0.004). Conclusions: In amateur MT fighters, an intervention with strength training and nutritional guidance is able to improve CAM through greater parasympathetic activation. Furthermore, there is an improvement in performance and body composition after the intervention.

## 1. Introduction

The popularity of martial arts in modern times is evident both as a competitive sport and as a form of exercise and physical conditioning [1]. Using either the entire body or a single fighting tool, such as a sword, martial arts allow fighters to employ a variety of different types of fighting techniques [1]. In recent decades, there has been a shift in the standards of martial arts, moving from negative connotations against violence to a perception as an activity of self-improvement, physical fitness, recreation, and logical competition [2]. Among the sports modalities of martial arts is Muay Thai (MT), which is characterized as a type of fighting that involves standing strikes with the combined use of fists, elbows, knees, shins, and feet [3]. MT is associated with good physical preparation, which makes it a very efficient full-contact fight for aerobic fitness both acutely and after training [4,5]. As with MT, the importance of training has been demonstrated in other martial arts. In judo, a specific training program can increase the ability to support greater weights and power during fights [6]. In karate, training increases power in the loaded countermovement jump exercise and maximum repetition strength in the squat exercise [7]. An important measure of the ability to perform high-intensity intermittent efforts is the 10 s frequency speed of kick test (FSKT-10s) and the multiple frequency speed of kick test (FSKT-mult). Although these tests have not been previously evaluated in MT fighters, they have been shown to be important performance tools in Taekwondo fighters and are associated with muscle mass and lower limb performance after training [8,9].

Heart rate (HR) is a measure of cardiac activity and varies according to several factors, including age, fitness level, and type of physical activity [10]. The autonomic nervous system (ANS) has a fundamental function in cardiac autonomic modulation (CAM), and its performance can be evaluated by analyzing CAM, which reflects sympathetic or parasympathetic modulations [11]. Physical activity causes changes in the nervous system, ranging from stimulation at the central level to the action of baroreceptors and the reflection of nervous activity in the muscles [10]. A critical factor in promoting CAM benefits is the intensity of training, which can be achieved through martial arts [12]. Importantly, low CAM has been associated with poorer performance and the development of chronic cardiovascular disease and higher prevalence of mortality [13,14]. Indeed, neural adaptations to physical training, as occurs in MT fighters, may increase resting parasympathetic nervous system (PNS) activation and decrease sympathetic nervous system (SNS) activation; this increases cardiovascular fitness, and therefore assessment of heart rate variability (HRV) in fighters may be of interest in sports medicine [15]. In this sense, it is important to look at parameters that represent vagal activation when measuring HRV, such as the square root of the mean squared differences of consecutive RR intervals (rMSSD), the percentage of adjacent RR intervals with a difference in duration greater than 50 ms (pNN50), the high frequency range (HF), and the standard deviation measuring the dispersion of points in the plot perpendicular to the line of identity (SD1).

Martial arts schools not only teach MT fighting techniques, but are also training centers where practitioners receive nutritional advice under the supervision of trainers [1]. Indeed, nutrition is an important aspect of MT as it can affect the performance, recovery, and overall health of fighters. To this end, it is essential to have a correct intake of not only fluids, but also carbohydrates, proteins, and fats [16]. In MT, some nutritional supplements may be useful, such as whey protein, which promotes muscle recovery/growth, creatine, which increases muscle strength/endurance, and beta-alanine, which increases endurance and the ability to train with intensity [17]. In addition to adequate nutritional status, nutrient intake is important because MT competitors are usually paired based on key characteristics, including body mass. In this sense, official weigh-ins are conducted prior to each competition to verify that the athlete’s body mass is consistent with their chosen weight class [18]. A recent study in mixed martial arts fighters showed difficulty in achieving adequate nutrient intake by category and recommended that these fighters receive attention regarding nutritional strategies [19].

It is unclear whether an intervention program based on strength training and nutritional counseling in MT fighters can provide CAM and body composition benefits in this population, although physical activity is essential to improve cardiovascular function. In this sense, the assessment of CAM by indirect measures, such as HRV, has gained popularity in the evaluation of martial arts fighters to provide information on cardiac regulation as well as neural adaptations to training [15]. Therefore, the effects of training in MT fighters to increase HRV, i.e., to increase parasympathetic modulation, should be investigated. Similarly, dietary habits are closely related to physical performance, although there is a gap in the literature on amateur or professional MT fighters regarding this aspect [19]. We hypothesized that strength training along with nutritional guidance can not only improve the performance of MT fighters, but can also activate the PNS and improve the nutritional profile. Therefore, the present study aimed to evaluate the effects of an 8-week strength training and nutritional counseling intervention on performance, CAM, and nutritional status.

## 2. Materials and Methods

### 2.1. Study Design

This was a longitudinal and interventional study. It included the results of baseline assessment and an intervention study.

### 2.2. Participants

Between March and November 2024, male amateur Muay Thai fighters from the AAZIZ Academy, Curitiba, Brazil, were evaluated. The following inclusion criteria were used: age ≥ 18 years and a minimum training period of 24 weeks. The following exclusion criteria were used: individuals with a history of cardiac arrhythmia or arterial hypertension, individuals with severe orthopedic disease that prevented them from practicing martial arts, use of alcohol or drugs, and inability to perform the performance tests.

The study was approved by the Research Ethics Committee of the Centro Universitário Augusto Motta (UNISUAM) (CA-AE-77325224.4.0000.5235; 14 March 2024). This trial was registered at ClinicalTrials.gov (NCT06338501, 29 March 2024). Informed consent was obtained from all subjects enrolled in the study.

### 2.3. Training Program and Intervention

A standard 8-week training program was performed in conjunction with a specific conditioning and strength training program based on the specifics of MT. These programs were performed three times per week and lasted 90 min. In each training session, the participant performed the following sequence: (1) stretching for 15 min; (2) warm-up for 10 min; (3) specific martial arts training for 1 h; and cool-down with stretching for 5 min. The MT training consisted of kicks, knees, punches, evasions, and defenses. Finally, kicks, knees, and defenses were performed both at rest and in motion. In this study, load control and interval adjustments between sets were used with 3 days of the whole body method, combining combat specific exercises with functional exercises [3,17]. Of note, the training program of these amateur fighters prior to the intervention consisted only of MT training, without any type of resistance exercise beyond the training itself and the physical conditioning performed during the sessions, such as push-ups, burpees, and jumping rope.

A nutritional protocol based on the individual needs of each participant was also used. One of the central points of this protocol was the diet, which should be based on a variety of healthy foods such as fruits, vegetables, whole grains, lean proteins and healthy fats according to the food pyramid. In this sense, nutritional guidance was provided on the nutritional adjustments that the participant should make, as well as metabolic tracking before and after training [20]. The following dietary suggestion was given with instructions to eat to satiety: (1) Breakfast: oatmeal with fresh fruit (e.g., five strawberries or a banana), an egg-white omelette with spinach and tomato, and a glass of fresh orange juice; (2) Mid-morning snack: a handful of walnuts or almonds, and a fruit (e.g., an apple or a pear); (3) Lunch: grilled chicken breast with a serving of white rice, steamed vegetables (cauliflower, carrots, and pear), and a salad with green vegetables, tomato and avocado dressed with olive oil and lemon; (4) Afternoon snack: a handful of Brazil nuts; (5) Dinner: white rice or a small serving of brown rice; and (6) Evening snack (optional, if needed): a glass of whole milk or plain yoghurt [21].

No amateur MT fighter consumed any type of supplement, and since it was only a dietary suggestion, compliance was not monitored. In addition, no fighter requested food substitutions, as these foods are commonly found in Brazilian cuisine [21]. Notably, we asked about the fighters’ dietary habits before the intervention, and none of them had previously received dietary advice or reported symptoms of malnutrition.

### 2.4. Measures

#### 2.4.1. 10 s Frequency Speed of Kick Test (FSKT-10s)

First, the participant stood 90 cm away from the bag. Then the participant started the FSKT-10s using a technique called bandal tchagui. After the sound signal, the participant performed as many kicks as possible, alternating between the right and left leg. The intensity was controlled by the coach’s instruction to the participant (maximum effort). The number of blows delivered was recorded in blows per minute [21].

#### 2.4.2. Multiple Frequency Speed of Kick Test (FSKT-Mult)

After a 1-min rest, the participant began the FSKT-mult test. This test consists of 5 sets of 10 s separated by 10 s of passive recovery. In each 10-set, the number of kicks was counted and added at the end of the sequence [22,23]. Performance was determined by the total number of kicks, i.e., the sum of the number of kicks in each set.

#### 2.4.3. Bioimpedance Analysis (BIA)

Prior to the performance tests, BIA was performed with a whole-body tetrapolar device (Sanny^®^, BIA 1010, São Paulo, Brazil) using an electrical frequency of 50 kHz. Prior to the test, the athlete was instructed to fast for 4 h and avoid physical activity for 12 h. In addition, the athlete was asked to remove any metal objects he/she was wearing. The participant was then placed in the supine position for electrode insertion. Four disposable electrodes were used: (1) on the hand, just below the third joint of the middle finger; (2) in the wrist region, between the styloid processes of the ulna and radius; (3) in the region below the second and third toes; and (4) on the ankle, in the central region between the lateral and medial malleoli. Basal metabolic rate (BMR) values were estimates by BIA [24].

#### 2.4.4. Cardiac Autonomic Modulation (CAM)

HRV measurements were performed between 6 pm and 8 pm to avoid daily variations. To assess CAM, participants were instructed to eat only a light meal and not to consume alcohol or stimulants such as coffee, tea, and chocolate for at least 12 h prior to the assessment so that there would be no direct influence on CAM at the time of collection. They were also asked to avoid high-intensity physical exercise for at least 2 h before the recording session [25]. CAM was assessed using a heart rate monitor (V800, Polar Electro Oy, Kempele, Finland). Measurements were taken at rest before the physical test, with the participant lying on a stretcher for 10 min, and during the physical test for 4 min. No warm-up was performed before the FSKT test. The RR interval signals were obtained from the heart rate monitor. These signals were exported to Kubios HRV software version 3.5 (Biosignal Analysis and Medical Image Group, Department of Physics, University of Kuopio, Finland) for HRV analysis using time domain, frequency domain, and Poincaré plot nonlinear analysis measures. The selected segments were the 5-min rest period and the visually most stable 2-min bout period. In this last analysis, the first 40 s of the bout and the last 30 s were excluded to avoid the vagal withdrawal period and the influence of blood pressure measurement, respectively. For HRV analysis, the data collected by the HR monitor were visually inspected for noise, and narrow peaks (<100 ms) were removed by linear interpolation. All signals were filtered using low-pass filters with a cutoff frequency of 20 Hz. Time domain analysis measures were as follows: (1) mean RR intervals (RR mean); (2) maximum HR; (3) standard deviation of RR intervals (SDNN), which captures overall HRV and reflects circadian heart rhythm; (4) rMSSD; (5) pNN50; and (6) triangular interpolation of RR interval histogram (TINN), which represents global autonomic activity. Frequency domain analysis measures were as follows: (1) the low frequency range (LF), which represents an index of SNS activity; (2) HF; and (3) LF/HF ratio, which is an indicator of sympathetic-vagal balance, with an increase possibly related to sympathetic predominance and a decrease indicating greater parasympathetic modulation. The power of the LF and HF components was evaluated in normalized units (nu). Finally, the following nonlinear Poincaré plot measures were evaluated: (1) SD1; (2) the standard deviation measuring the dispersion of points along the line of identity (SD2), which represents global HRV variability; (3) the ratio SD2/SD1, which represents PNS action; and (4) the approximate entropy (ApEn), which takes into account the complex dynamics of biological systems in series of RR intervals, where ApEn values close to 0 are considered highly regular and higher values imply greater complexity. We also evaluated the PNS index, calculated in the Kubios HRV software from measures of RR interval mean, rMSSD, and Poincaré plot index SD1, and the SNS index, calculated in the Kubios HRV software from measures of mean HR, a geometric measure of HRV reflecting cardiovascular system load, and Poincaré plot index SD2. Previously published recommendations [26] were followed.

### 2.5. Statistical Analysis

Histograms and the Shapiro-Wilk test were used for the evaluation of the data distribution. Data were expressed using measures of central tendency and dispersion appropriate for numerical data. Inferential analysis consisted of the Student’s *t*-test for paired samples or the Wilcoxon signed-rank test to assess the variation between the pre- and post-intervention moments in the resting and physical test conditions. The 5% level was used as the criterion for determining significance. Statistical analysis was performed using SPSS version 26 (IBM Corp., Armonk, NY, USA).

## 3. Results

Of the 34 male MT fighters eligible for the study, 13 were excluded because they did not return for the post-intervention assessment. The mean age of the participants was 29.2 ± 8.1 years. The median number of kicks on the FSKT-10 was 20 (16–24) and 30 (20–34) at pre- and post-intervention, respectively (*p* = 0.0008). The median number of kicks on the FSKT-mult was 92 (80.5–111) and 108 (92–134) at pre- and post-intervention, respectively (*p* = 0.032). In BIA, there was a significant increase in both fat-free mass (FFM, 68 (62–85) vs. 71 (62–84) kg, *p* = 0.031) and BMR as determined by BIA (1878 (1748–2125) vs. 2063 (1806–2414 kcal), *p* = 0.020) between pre- and post-intervention measurements. Comparisons of body composition between pre- and post-intervention are shown in Table 1.

When comparing HRV indices obtained before physical test (10 min) between pre- and post-intervention, an increase in HF [26.6 (23.2–34.8) vs. 78 (62.9–82) ms, *p* < 0.0001] and SD1 [28.9 (22.9–36.8) vs. 53.4 (40–77.8) ms, *p* = 0.001] was observed. There was a trend towards an increase in the LF/HF ratio [1.47 (0.73–2.69) vs. 1.07 (0.61–1.27), *p* = 0.073]. Comparisons of resting HRV indices obtained before physical test between pre- and post-intervention are shown in Table 2 and Figure 1.

When comparing the HRV indices obtained during the physical test (4 min) between pre- and post-intervention, significant increases were observed in the following variables: rMSSD [55 (27–76) vs. 79 (47–131) ms, *p* = 0. 005]; pNN50 [9.6 (5–26.1) vs. 36.2 (24.4–48)%, *p* = 0.002]; HF [19.5 (16.9–22.5) vs. 59.5 (51.5–65.6) nu, *p* < 0.0001]; and SD1 [50.4 (39.4–79.5) vs. 84.2 (74.8–88.1) ms, *p* = 0.004]. Comparisons of HRV indices obtained during the physical test between pre- and post-intervention are shown in Table 3 and Figure 2.

## 4. Discussion

Based on anthropometric data and physiological profile (e.g., limb strength and cardiovascular fitness), martial arts competitors are classified into categories [27]. Thus, understanding the changes that occur in body composition and CAM after an intervention program with strength training and nutritional guidance is essential to assess performance and adequately guide exercise prescription. The main findings of the present study were that, after an 8-week intervention in amateur MT fighters, there was an improvement in performance assessed by an increase in the number of strikes applied. In these fighters, there was an improvement in body composition assessed by FFM. Furthermore, the CAM assessed between the 2 pre-combat rest periods (pre- and post-intervention) showed a vagal withdrawal assessed by the elevation of HF and SD1. This parasympathetic activation became more evident when the 2 exercise periods were compared (pre- and post-intervention), with an increase in rMSSD, pNN50, HF and SD1. To our knowledge, this is the first study that evaluated CAM in MT fighters after an intervention based on strength training and nutritional guidance.

Kicking is a fundamental aspect of martial arts such as MT, where the impact and speed of the kick depends on numerous variables, including body flexibility and lower limb strength [28]. Using the FSKT-10s and FSKT-mult techniques, which are among the most frequently used techniques in official competition [29], we observed a significant increase in the number of strikes applied in both FSKT-10s and FSKT-mult. Proposed mechanisms for determining kick velocity and impact force include: (1) effective use of body mass; (2) use of proximal to distal lower limb movement; (3) adequate coordination; and (4) muscle activation. There are a number of conditions that influence kick performance, such as flexibility, lower limb strength, hip muscle strength, and jumping ability [30]. Therefore, we believe that an 8-week intervention in MT fighters should be promoted for MT fighters because lower body strength is likely to exert its effect by increasing the fighter’s ability to generate ground reaction forces. However, to increase kick effectiveness, not only ground reaction force but also impact force and foot speed are required [31]. It is worth noting that although the FSKT has been used almost exclusively to assess physical performance in taekwondo fighters [22,29], our findings point to its importance for assessing physical performance and CAM in MT fighters as well.

In MMA fighters, Anyżewska and colleagues [32] reported insufficient energy intake from carbohydrates, as well as decreased minerals (iodine, potassium, calcium) and vitamins (D, folate, C, E) throughout a training day. Using a nutritional protocol based on the individual needs of each athlete, we observed an increase in FFM and BMR. In line with our findings, Cha and Jee [33] showed that Wushu Nanquan training—which is also another type of martial art—is effective not only in improving cardiac function, but also in improving body composition, which is accompanied by an increase in BMR. In this sense, it is worth highlighting the debate about rapid weight loss (RWL) and rapid weight gain (RWG). Despite the potential health and performance risks associated with RWL, many fighters believe that RWL followed by RWG provides a competitive advantage. Interestingly, a recent study showed that MT competition winners have greater RWL and RWG than losers, and rapid weight change in women appears to be associated with competitive success in this group [18]. It is worth noting that in our study there was no consumption of supplements or use of doping substances such as testosterone by the fighters, according to the initial pre-participation and final informal investigations. However, this opens perspectives for future research on the use of supplements such as creatine and beta-alanine on FSKT performance and ANS function.

In the present study, we assessed HRV using both linear methods to quantify sympathovagal balance and nonlinear methods to assess the complexity of the interaction of biological systems in the heart [34]. In the present study, we observed greater vagal activation after the intervention. Interestingly, this greater PNS activity was more pronounced when comparing the combat periods (rMSSD, pNN50, HF, SD1) with the pre-combat rest periods (HF and SD1). Interval or intermittent training has been shown to promote improvements in CAM, especially at higher intensities [35]. In contrast to our findings, Suetake and colleagues [10] observed no improvement in CAM among MT fighters and healthy controls after nine months of intervention. Interestingly, these authors observed significant increases in rMSSD and SD1 post-intervention only in judo fighters. However, it is worth noting that Suatake and collegues [10] used only training as part of the intervention, without any nutritional guidance. Thus, we believe that the assessment of CAM before and after intervention may be an important parameter for monitoring cardiovascular health, especially when training is associated with nutritional guidance.

The strength of this study is that it demonstrated important effects on performance, CAM, and nutritional status following an 8-week protocol of strength training and nutritional counseling in amateur MT fighters. However, several limitations should be highlighted. First, the sample is relatively small and there is no control group for either habitual physical activity or dietary intake. Second, martial arts have some characteristics, such as the use of fast and explosive movements, physical confrontation, and frequent breaks, which make it difficult to monitor HRV during the fight [36]. Third, the nutritional protocol, although based on the individual needs of each athlete, was not monitored; however, this is a condition that is carried out in real life. Despite these limitations, our study can serve as a starting point for randomized controlled trials with a larger number of fighters from different modalities, with the application of long-term intervention.

## 5. Conclusions

In amateur MT fighters, an 8-week intervention of strength training and nutritional counseling is able to improve CAM, particularly through parasympathetic activation. This greater PNS activity is better seen in HRV measurements taken during competition than during rest before competition. In these fighters, there is a better performance after the intervention as assessed by the FSKT. In addition, there is an improvement in body composition as indicated by an increase in both FFM and BMR. Based on these results, the use of an 8-week intervention is highly recommended for amateur MT fighters, and this should be kept in mind by coaches, physical trainers, and nutritionists of this type of martial art.

## Figures and Tables

**Figure 1 sports-13-00072-f001:**
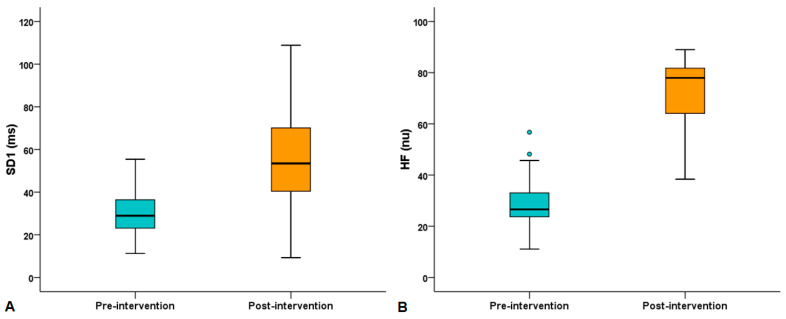
Comparisons of (**A**) the high frequency range (HF, *p* < 0.0001) and (**B**) the standard deviation measuring the dispersion of points in the plot perpendicular to the line of identity (SD1, *p* = 0.001) obtained at rest before physical test between pre- and post-intervention. Data represent median (quartiles). Circles below or above the box plot indicate outliers.

**Figure 2 sports-13-00072-f002:**
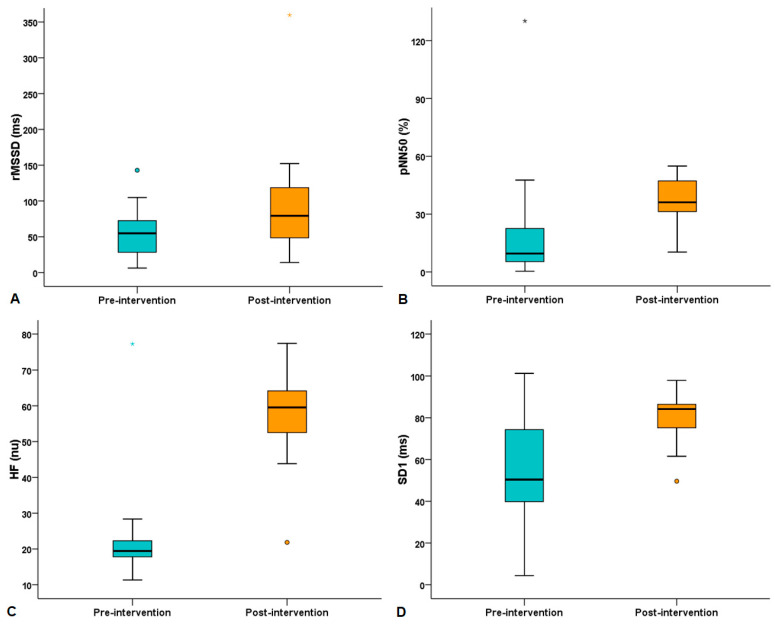
Comparisons of (**A**) square root of the mean squared differences of successive RR intervals (rMSSD, *p* = 0.005), (**B**) the percentage of adjacent RR intervals with a difference in duration greater than 50 ms (pNN50, *p* = 0.002), (**C**) the high frequency range (HF, *p* < 0.0001), and (**D**) the standard deviation measuring the dispersion of points in the plot perpendicular to the line of identity (SD1, *p* = 0.004) obtained during the physical test between pre- and post-intervention. Data represent median (quartiles). Circles or stars below or above the box plot indicate outliers.

**Table 1 sports-13-00072-t001:** Body composition comparisons between pre- and post-intervention.

Variables	Pre-Training	Post-Training	*p*-Value
Body mass (kg)	83.4 ± 18	84.2 ± 16.7	0.28
BMI (kg/m^2^)	26.3 ± 4.6	26.6 ± 4	0.20
Body fat (%)	13 (10–14)	12 (11–13)	0.89
Body fat (kg)	10.1 (7.5–13.3)	9.9 (7.4–12.9)	0.81
FFM (%)	87 (78–89)	88 (86–90)	0.088
FFM (kg)	68 (62–85)	71 (62–84)	**0.031**
TBW (%)	64 (64–65)	64 (63–66)	0.68
TBW (L)	50 (45–63)	52 (46–60)	0.27
BMR (kcal)	1878 (1748–2125)	2063 (1806–2414)	**0.020**

BMI: body mass index; FFM: fat-free mass; TBW: total body water; and BMR: basal metabolic rate. Data represent mean ± standard deviation or median (quartiles). The values in bold refer to significant differences.

**Table 2 sports-13-00072-t002:** Comparisons of resting heart rate variability indices before physical test between pre- and post-intervention.

Variables	Pre-Training	Post-Training	*p*-Value
RR mean (ms)	765 (662–891)	810 (738–900)	0.57
Maximum HR (bpm)	78 (67–91)	74 (66.5–81.5)	0.63
SDNN (ms)	80 (62–106)	87 (57–254)	0.34
rMSSD (ms)	78 (60–125)	87 (62–323)	0.41
pNN50 (%)	30.9 (10.3–40.1)	34.5 (30.5–48.3)	0.11
TINN (ms)	546 (409–966)	537 (333–2376)	0.43
LF (nu)	51.8 (37.7–55.9)	49.5 (32.2–52.8)	0.09
HF (nu)	26.6 (23.2–34.8)	78 (62.9–82)	**<0.0001**
LF/HF	1.47 (0.73–2.69)	1.07 (0.61–1.27)	0.073
SD1 (ms)	28.9 (22.9–36.8)	53.4 (40–77.8)	**0.001**
SD2 (ms)	90 (73–121)	102 (67–276)	0.26
SD2/SD1	1.38 (1.18–1.77)	1.61 (1.32–2.12)	0.29
ApEn	61 (0.85–134)	125 (0.77–229)	0.41
PNS index	0.71 (−0.56–2.27)	0.82 (−0.33–7.13)	0.54
SNS index	0.37 (738–900)	−0.05 (−0.87–0.95)	0.57

Data represent median (quartiles). The values in bold refer to significant differences. RR mean: mean RR intervals; HR: heart rate; SDNN: standard deviation of RR intervals; rMSSD: the square root of the mean squared differences of consecutive RR intervals; pNN50: the percentage of adjacent RR intervals with a difference in duration greater than 50 ms; TINN: triangular interpolation of RR interval histogram; LF: the low frequency range; HF: the high frequency range; SD1: the standard deviation measuring the dispersion of points in the plot perpendicular to the line of identity; SD2: the standard deviation measuring the dispersion of points along the line of identity; ApEn: approximate entropy; PNS, parasympathetic nervous system; SNS: sympathetic nervous system.

**Table 3 sports-13-00072-t003:** Comparisons of heart rate variability indices obtained during the physical test between pre- and post-intervention.

Variables	Pre-Training	Post-Training	*p*-Value
RR mean (ms)	403 (382–452)	387 (379–420)	0.48
Maximum HR (bpm)	149 (133–157)	155 (143–158.5)	0.41
SDNN (ms)	71 (45–128)	46 (38–82)	0.60
rMSSD (ms)	55 (27–76)	79 (47–131)	**0.005**
pNN50 (%)	9.6 (5–26.1)	36.2 (24.4–48)	**0.002**
TINN (ms)	539 (357–722)	431 (286–708)	0.74
LF (nu)	55.8 (33.2–69.3)	39 (29.6–67.2)	0.36
HF (nu)	19.5 (16.9–22.5)	59.5 (51.5–65.6)	**<0.0001**
LF/HF	1.26 (0.50–2.07)	0.64 (0.42–2.20)	0.45
SD1 (ms)	50.4 (39.4–79.5)	84.2 (74.8–88.1)	**0.004**
SD2 (ms)	50 (41–85)	72 (44–146)	0.69
SD2/SD1	1.13 (1–1.27)	1.17 (1–1.74)	0.54
ApEn	0.32 (0.27–0.50)	0.44 (0.34–0.71)	0.071
PNS index	−2.38 (−2.81–0.82)	−0.77 (−2.11–1.03)	0.16
SNS index	7.66 (5.38–8.56)	6.75 (4.87–7.81)	0.61

Data represent median (quartiles). The values in bold refer to significant differences. RR mean: mean RR intervals; HR: heart rate; SDNN: standard deviation of RR intervals; rMSSD: the square root of the mean squared differences of consecutive RR intervals; pNN50: the percentage of adjacent RR intervals with a difference in duration greater than 50 ms; TINN: triangular interpolation of RR interval histogram; LF: low frequency range; HF: the high frequency range; SD1: the standard deviation measuring the dispersion of points in the plot perpendicular to the line of identity; SD2: the standard deviation measuring the dispersion of points along the line of identity; ApEn: approximate entropy; PNS, parasympathetic nervous system; SNS: sympathetic nervous system.

## Data Availability

The data supporting the conclusions of this article can be made available by the authors upon reasonable request.

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
