# Peer review of "Adaptations of the Autonomic Nervous System and Body Composition After 8 Weeks of Specific Training and Nutritional Re-Education in Amateur Muay Thai Fighters: A Clinical Trial"

_sports, 2025, doi:10.3390/sports13030072_

Round 1
Reviewer 1 Report
Comments and Suggestions for Authors
Thank you for the opportunity to evaluate this scientific article. In an effort to synthesize, I will indicate my suggestions for correcting the informational content for each chapter, as follows:
Introduction:
It would be necessary to clarify the research hypothesis in the sense that it more clearly illustrates the expectations of the study.
Also, the relevance of Muay Thai training compared to other combat sports can be more clearly highlighted.
In Materials and Methods:
More detailed explanation of the nutritional protocol: Were generic foods indicated (fruits, vegetables, whole grains, etc.) but were supplements also used? If so, which ones, synthetic or natural products. Was the daily intake monitored? In addition, when we mention fruits and vegetables, another discussion can arise here... organic products that are really of quality or poor quality fruits and vegetables full of pesticides?
Regarding supplements, was there a check whether or not substances that could fall under the incidence of doping were used?
To Results:
In principle I would leave the informational content as it is, maybe aspects should be added that come from what I mentioned previously, if applicable.
To Discussion:
Exploring practical implications in the sense that how could coaches and nutritionists apply the conclusions of this study in practice?
Mentioning the limitations at the end of this chapter in a more explicit way.
Conclusions and Bibliography - no comments

Author Response
Response to Reviewer 1 Comments
First, we would like to thank you for your time and comments, which were very useful. We agree that some points in the initial manuscript needed more detail. Below, we reply to each of your comments. The text modifications are highlighted.
Thank you for the opportunity to evaluate this scientific article. In an effort to synthesize, I will indicate my suggestions for correcting the informational content for each chapter, as follows:
Introduction:
It would be necessary to clarify the research hypothesis in the sense that it more clearly illustrates the expectations of the study.
Response: As requested, we clarify the research hypothesis in the sense that it warmly illustrates the expectations of the study as follows: “We hypothesized that strength training along with nutritional guidance can not only improve the performance of MT fighters, but can also activate the PNS and improve the nutritional profile.”
Also, the relevance of Muay Thai training compared to other combat sports can be more clearly highlighted.
Response: As requested, we have pointed out the relevance of Muay Thai training compared to other combat sports as follows: “As with MT, the importance of training has been demonstrated in other martial arts. In judo, a specific training program can increase the ability to support greater weights and power during fights [6]. In karate, training increases power in the loaded countermove-ment jump exercise and maximum repetition strength in the squat exercise [7].”
In Materials and Methods:
More detailed explanation of the nutritional protocol: Were generic foods indicated (fruits, vegetables, whole grains, etc.) but were supplements also used? If so, which ones, synthetic or natural products. Was the daily intake monitored? In addition, when we mention fruits and vegetables, another discussion can arise here... organic products that are really of quality or poor quality fruits and vegetables full of pesticides?
Response: Thank you for your comments. In response to your requests, the following sentences have been added to the Methods section of the manuscript: “No amateur MT fighter consumed any type of supplement, and since it was only a dietary suggestion, compliance was not monitored. In addition, no fighter requested food substitutions, as these foods are commonly found in Brazilian cuisine [21]. Notably, we asked about the fighters' dietary habits before the intervention, and none of them had pre-viously received dietary advice or reported symptoms of malnutrition.”
Regarding supplements, was there a check whether or not substances that could fall under the incidence of doping were used?
Response: This answer is commented on above. In addition, we have added the following sentences to the Discussion section: “It is worth noting that in our study there was no consumption of supplements or use of doping substances such as testosterone by the fighters, according to the initial pre-participation and final informal investigations. However, this opens perspectives for future research on the use of supplements such as creatine and beta-alanine on FSKT per-formance and ANS function.”
To Results:
In principle I would leave the informational content as it is, maybe aspects should be added that come from what I mentioned previously, if applicable.
Response: Thank you for your comment. The information requested above has been added to the Methods section.
To Discussion:
Exploring practical implications in the sense that how could coaches and nutritionists apply the conclusions of this study in practice?
Response: Thank you for your suggestion. We have added some insights into the practical implications of how coaches and nutritionists can apply the findings of this study throughout the Discussion section.
Mentioning the limitations at the end of this chapter in a more explicit way.
Response: As required, the limitations of the study were mentioned at the end of the Discussion section in a more explicit way.
Conclusions and Bibliography - no comments
Response: The authors would like to thank you once again for your comments, which have contributed greatly to the improvement of the manuscript.

Reviewer 2 Report
Comments and Suggestions for Authors
Abstract
Lines 15-16: Even though the information presented is accurate, I am not sure how cardiovascular mortality is associated with Muay Thai athletes. Considering the aim of the study, the first sentence of the abstract is irrelevant.
Be consistent with the use of fighters or athletes. Use one of the two for consistency.
Introduction
Line 63: The statement is accurate, but that doesn’t justify the need to assess HRV in athletes.
The introduction lacks several important parameters and does not justify the need for this study. The authors failed to mention the parameters that they are going to assess. They should have been more specific as to which parameters of HRV (SNDD, rMSSD, pNN50, LF, HF) they are going to assess and why. Also, how would the strength training program that the authors are going to use affect FSKT-10 sec or FSKT mult? What was previously done? In general, the introduction needs to be revised and developed in a way to show the need for this study.
Lines 79-80: are you assessing your intervention or the effects of MT on body composition and CAM?
Methods
What was the training program of these athletes before?
How did you obtain their nutritional habits before the intervention?
How did you prepare the athletes for the resting HRV measurements? Considering the great variability of those measurements, did you have some baseline assessment? For how long did you record the HRV? Some of those frequency domains, to be accurate, need to be recorded for much longer than others, such as the SDNN.
In the results, you have reported some basal metabolic rate measurements. How did you get those? Did you use direct calorimetry or are you reporting estimated values from BIA?
Finally, you need to provide enough information regarding the nutritional intervention. You cannot suggest that the nutritional intervention worked without providing enough information for someone to replicate your intervention.
Results
You need to justify for how long you recorded HRV. Also, how did you get the basal metabolic rate? Also, for the Tables, you need to explain that you are presenting the mean and the range in the parenthesis.
Discussion
Again, the comparison of your results to studies that included the elderly or individuals with chronic obstructive lung disease is not appropriate. I would recommend comparing your results to the athletic population or healthy young individuals.

Author Response
Response to Reviewer 2 Comments
First, we would like to thank you for your time and comments, which were very useful. We agree that some points in the initial manuscript needed more detail. Below, we reply to each of your comments. The text modifications are highlighted.
Abstract
Lines 15-16: Even though the information presented is accurate, I am not sure how cardiovascular mortality is associated with Muay Thai athletes. Considering the aim of the study, the first sentence of the abstract is irrelevant.
Response: Thank you for your comments. The first sentence of the abstract (and also in the manuscript text) has been changed to align with the aim of the study as follows: “Considering that the nervous system regulates cardiac autonomic modulation (CAM) and that low CAM is associated with poorer performance, it is essential to evaluate the effects of training to increase parasympathetic modulation in Muay Thai (MT) fighters.”
Be consistent with the use of fighters or athletes. Use one of the two for consistency.
Response: Thank you for your observation. We adopt it the use of Fighters throughout the manuscript.
Introduction
Line 63: The statement is accurate, but that doesn’t justify the need to assess HRV in athletes.
Response: Thank you for your comment. To emphasize the importance of assessing HRV in athletes whenever possible, the sentence has been changed to read as follows: “Indeed, neural adaptations to physical training, as occurs in MT fighters, may increase resting parasympathetic nervous system (PNS) activation and decrease sympathetic nervous system (SNS) activation; this increases cardiovascular fitness, and therefore as-sessment of heart rate variability (HRV) in fighters may be of interest in sports medicine [15].”
The introduction lacks several important parameters and does not justify the need for this study. The authors failed to mention the parameters that they are going to assess. They should have been more specific as to which parameters of HRV (SNDD, rMSSD, pNN50, LF, HF) they are going to assess and why. Also, how would the strength training program that the authors are going to use affect FSKT-10 sec or FSKT mult? What was previously done? In general, the introduction needs to be revised and developed in a way to show the need for this study.
Response: Thank you for your comments. We have added in the Introduction section which HRV parameters we evaluated and why. In addition, we have highlighted the FSKT-10 sec or the FSKT mult in the Introduction section to show their importance in other studies as follows:
- “Indeed, neural adaptations to physical training, as occurs in MT fighters, may increase resting parasympathetic nervous system (PNS) activation and decrease sympathetic nervous system (SNS) activation; this increases cardiovascular fitness, and therefore as-sessment of heart rate variability (HRV) in fighters may be of interest in sports medicine [15]. In this sense, it is important to look at parameters that represent vagal activation when measuring HRV, such as the square root of the mean squared differences of consec-utive RR intervals (rMSSD), the percentage of adjacent RR intervals with a difference in duration greater than 50 ms (pNN50), the high frequency range (HF), and the standard deviation measuring the dispersion of points in the plot perpendicular to the line of iden-tity (SD1).”
- “An important measure of the ability to perform high-intensity intermittent efforts is the 10 s frequency speed of kick test (FSKT-10s) and the multiple frequency speed of kick test (FSKT-mult). Although these tests have not been previously evaluated in MT fighters, they have been shown to be important performance tools in Taekwondo fighters and are asso-ciated with muscle mass and lower limb performance after training [8,9].”
Lines 79-80: are you assessing your intervention or the effects of MT on body composition and CAM?
Response: Thank you for your comment. In fact, we were evaluating the effects of the intervention, not the effects of Muay Thai. Therefore, the sentence has been changed as follows: “It is unclear whether an intervention program based on strength training and nutritional counseling in MT fighters can provide CAM and body composition benefits in this popu-lation, although physical activity is essential to improve cardiovascular function.”
Methods
What was the training program of these athletes before?
Response: As requested, the following sentence was added to the Methods section of the manuscript: “Of note, the training program of these amateur fighters prior to the intervention consisted only of MT training, without any type of resistance exercise beyond the training itself and the physical conditioning performed during the sessions, such as push-ups, burpees, and jumping rope.”
How did you obtain their nutritional habits before the intervention?
Response: As requested, the following sentence was added to the Methods section of the manuscript: “Notably, we asked about the fighters' dietary habits before the intervention, and none of them had previously received dietary advice or reported symptoms of malnutrition.”
How did you prepare the athletes for the resting HRV measurements? Considering the great variability of those measurements, did you have some baseline assessment? For how long did you record the HRV? Some of those frequency domains, to be accurate, need to be recorded for much longer than others, such as the SDNN.
Response: As requested, the preparation of the fighters for resting HRV measurements and other technical details were described in the new version of the manuscript as follows: “HRV measurements were performed between 6 pm and 8 pm to avoid daily varia-tions. To assess CAM, participants were instructed to eat only a light meal and not to consume alcohol or stimulants such as coffee, tea, and chocolate for at least 12 h prior to the assessment so that there would be no direct influence on CAM at the time of collection. They were also asked to avoid high-intensity physical exercise for at least 2 h before the recording session [25]. CAM was assessed using a heart rate monitor (V800, Polar Electro Oy, Finland). Measurements were taken at rest before the physical test, with the partici-pant lying on a stretcher for 10 minutes, and during the physical test for 4 minutes. No warm-up was performed before the FSKT test. The RR interval signals were obtained from the heart rate monitor. These signals were exported to Kubios HRV software (Biosignal Analysis and Medical Image Group, Department of Physics, University of Kuopio, Fin-land) for HRV analysis using time domain, frequency domain, and Poincaré plot nonlin-ear analysis measures. The selected segments were the 5-minute rest period and the visu-ally most stable 2-minute bout period. In this last analysis, the first 40 s of the bout and the last 30 s were excluded to avoid the vagal withdrawal period and the influence of blood pressure measurement, respectively. For HRV analysis, the data collected by the HR mon-itor were visually inspected for noise, and narrow peaks (<100 ms) were removed by linear interpolation. All signals were filtered using low-pass filters with a cutoff frequency of 20 Hz.”
In the results, you have reported some basal metabolic rate measurements. How did you get those? Did you use direct calorimetry or are you reporting estimated values from BIA?
Response: As requested, we have added the following sentence to the Methods section of the manuscript: “Basal metabolic rate (BMR) values were estimates by BIA.”
Finally, you need to provide enough information regarding the nutritional intervention. You cannot suggest that the nutritional intervention worked without providing enough information for someone to replicate your intervention.
Response: In response to your request, we have added the following information about the dietary intervention to the manuscript: “The following dietary suggestion was given with instructions to eat to satiety: (1) Break-fast: oatmeal with fresh fruit (e.g. five strawberries or a banana), an egg-white omelette with spinach and tomato, and a glass of fresh orange juice; (2) Mid-morning snack: a handful of walnuts or almonds, and a fruit (e.g. an apple or a pear); (3) Lunch: grilled chicken breast with a serving of white rice, steamed vegetables (cauliflower, carrots, and pear), and a salad with green vegetables, tomato and avocado dressed with olive oil and lemon; (4) Afternoon snack: a handful of Brazil nuts; (5) Dinner: white rice or a small serving of brown rice; and (6) Evening snack (optional, if needed): a glass of whole milk or plain yoghurt [21].”
You need to justify for how long you recorded HRV. Also, how did you get the basal metabolic rate? Also, for the Tables, you need to explain that you are presenting the mean and the range in the parenthesis.
Response: As required, changes have been made throughout the Results section. It is important to note that the data presented in tables and figures is shown as the median (quartiles), rather than the mean and standard deviation.
Discussion
Again, the comparison of your results to studies that included the elderly or individuals with chronic obstructive lung disease is not appropriate. I would recommend comparing your results to the athletic population or healthy young individuals.
Response: Thank you for your suggestion. We have removed comparisons of our results with studies that included elderly individuals or individuals with chronic obstructive pulmonary disease. The comparison was made with the only study in the literature that compared Muay Thai fighters, judo fighters, and healthy controls as follows: “In contrast to our findings, Suetake and colleagues [10] observed no improvement in CAM among MT fighters and healthy controls after nine months of intervention. Interestingly, these authors observed significant increases in rMSSD and SD1 post-intervention only in judo fighters. However, it is worth noting that Suatake and collegues [10] used only train-ing as part of the intervention, without any nutritional guidance. Thus, we believe that the assessment of CAM before and after intervention may be an important parameter for monitoring cardiovascular health, especially when training is associated with nutritional guidance.”

Round 2
Reviewer 1 Report
Comments and Suggestions for Authors
It is ok now.
Comments on the Quality of English LanguageIt is ok now.
Reviewer 2 Report
Comments and Suggestions for Authors
I am happy with the extensive changes. I find this work significantly improved and have no further requests.